# Collaborative Uncertainty
# in Multi-Agent Trajectory Forecasting

**Bohan Tang**[1]  **Yiqi Zhong**[2]  **Ulrich Neumann**[2]  **Gang Wang**[3]  **Ya Zhang**[1]  **Siheng Chen**[1]*

[1]Shanghai Jiao Tong University    [2]University of Southern California    [3]Beijing Institute of Technology

[1]tangbohan@alumni.sjtu.edu.cn   [1]{sihengc, ya_zhang}@sjtu.edu.cn

[2]{yiqizhon, uneumann}@usc.edu [3]gangwang@bit.edu.cn

## Abstract

Uncertainty modeling is critical in trajectory forecasting systems for both interpretation and safety reasons. To better predict the future trajectories of multiple agents, recent works have introduced interaction modules to capture interactions among agents. This approach leads to correlations among the predicted trajectories. However, the uncertainty brought by such correlations is neglected. To fill this gap, we propose a novel concept, *collaborative uncertainty* (CU), which models the uncertainty resulting from the interaction module. We build a general CU-based framework to make a prediction model learn the future trajectory and the corresponding uncertainty. The CU-based framework is integrated as a plugin module to current state-of-the-art (SOTA) systems and deployed in two special cases based on multivariate Gaussian and Laplace distributions. In each case, we conduct extensive experiments on two synthetic datasets and two public, large-scale benchmarks of trajectory forecasting. The results are promising: 1) The results of synthetic datasets show that CU-based framework allows the model to appropriately approximate the ground-truth distribution. 2) The results of trajectory forecasting benchmarks demonstrate that the CU-based framework steadily helps SOTA systems improve their performances. Specially, the proposed CU-based framework helps VectorNet improve by 57 cm regarding Final Displacement Error on nuScenes dataset. 3) The visualization results of CU illustrate that the value of CU is highly related to the amount of the interactive information among agents.

## 1 Introduction

A multi-agent trajectory forecasting system aims to predict future trajectories of multiple agents based on their observed trajectories and surroundings [1, 2]. Precise trajectory prediction provides essential information for decision making and safety in numerous intelligent systems, including autonomous vehicles [3, 4, 5, 6], drones [7], and industrial robotics [8, 9].

The rapid development of deep learning has enabled a number of deep-learning-based algorithms to handle multi-agent trajectory forecasting [3, 4, 5, 6, 10, 11, 12, 13, 14, 15]. These methods exhibit state-of-the-art performances, with some having been integrated into real-world systems. However, deep-learning-based forecasting is not always reliable or interpretable [16, 17, 18]. In circumstances when noises from the environment are overwhelmingly distracting, or when the situation has never been encountered before, a deep-learning-based algorithm could provide baffling predictions, which might cause terrible tragedies. A fundamental challenge is to know when we could rely on those deep-learning-based forecasting algorithms. To tackle this problem, one solution is to report the uncertainty of each prediction. Finding ways to best conceptualize and measure the prediction uncertainty of deep-learning-based algorithms becomes an imperative, which motivates this work.

---

*The corresponding author is Siheng Chen.

35th Conference on Neural Information Processing Systems (NeurIPS 2021).

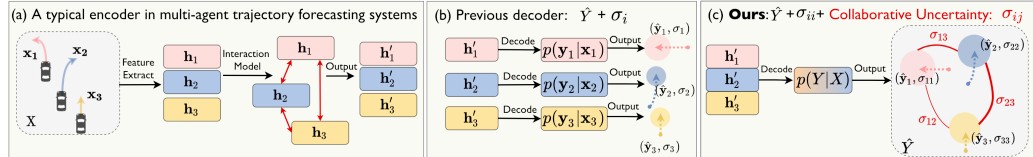

Figure 1: **Uncertainty modeling in multi-agent trajectory forecasting.** (a) a typical pipeline of an encoder in multi-agent trajectory forecasting systems. (b) and (c) illustrate the decoder pipeline of previous methods and our method respectively. Previous methods output the predicted trajectory $\hat{Y}$ and individual uncertainty $\sigma_i$, and our method additionally outputs *collaborative uncertainty* $\sigma_{ij}$.

There are two main types of uncertainty to model in deep-learning-based algorithms [19]: 1) aleatoric uncertainty, regarding information aside from statistic models, which data cannot explain; 2) epistemic uncertainty, the uncertainty inside a model, when the model lacks knowledge of the system/process being modeled (e.g., due to limited training data). As it is most effective to model aleatoric uncertainty in big data regimes such as those common to deep learning with image data [17], this work focuses on aleatoric uncertainty. In the following passage, we use the term "uncertainty" to represent aleatoric uncertainty. [16] uses the predictive variance to approximate uncertainty in the Bayesian deep learning model, which has been widely adapted in many works [20, 21, 22, 11] for uncertainty modeling in multi-agent trajectory forecasting. However, the predictive variance of a single agent alone may not suffice to reflect the complete landscape of uncertainty, especially when agent-wise interaction is present. Recent works that attempt to exploit the interaction among agents have impressively boosted the prediction precision, which further highlights the need to better measure uncertainty in multi-agent trajectory forecasting. We seek to build a more sophisticated and robust measurement for capturing the previously neglected uncertainty brought by correlated predictions.

In this paper, we coin a concept *individual uncertainty* (IU) to describe the uncertainty that can be approximated by the predictive variance of a single agent. Relatively, we propose a new concept, *collaborative uncertainty* (CU) to estimate the uncertainty resulting from the usage of interaction modules in prediction models. We further introduce an original probabilistic CU-based framework to measure both individual and collaborative uncertainty in the multi-agent trajectory forecasting task. We apply this framework to two special cases: multivariate Gaussian distribution and multivariate Laplace distribution. In each case, our CU-based framework allows our model to simultaneously learn the mappings that are from input data to 1) accurate prediction, 2) individual uncertainty, and 3) collaborative uncertainty; see Figure 1 for model illustration. Extensive experiments demonstrate that CU modeling yields significantly larger performance gains in prediction models equipped with interaction modules (See Figure 4), confirming that CU is highly related to the existence of the interaction modeling procedure, and adding CU modeling benefits accurate predictions.

The contributions of this work are summarized as follows:

• We propose, analyze, and visualize a novel concept, *collaborative uncertainty* (CU), to model the uncertainty brought by the interaction modules in multi-agent trajectory forecasting.

• We design a general CU-based framework to empower a prediction model to generate a probabilistic output, where the mean is the future trajectory and the covariance reflects the corresponding uncertainty. Under this framework, we show two special cases based on multivariate Gaussian and Laplace distributions respectively.

• We conduct extensive experiments to validate the CU-empowered prediction model on both synthetic datasets and two large-scale real-world datasets. On self-generated synthetic datasets, we validate the proposed method is able to closely reconstruct the ground-truth distribution. On the public benchmarks, the CU-empowered prediction model consistently outperforms the corresponding one without CU. Specially, by leveraging the proposed CU, VectorNet improves by 57 cm regarding Final Displacement Error (FDE) on nuScenes dataset!

## 2  Related Works

**Aleatoric uncertainty modeling in deep learning.** Recent efforts are rising as to improve the measurement of aleatoric uncertainty in deep learning models. One seminal work is [17]. It proposes a unified Bayesian deep learning framework to explicitly represent aleatoric uncertainty using predictive variance for generic regression and classification tasks. Many existing works [23, 24, 25, 26, 27, 28] follow this idea and formulate uncertainty as learned loss attenuation. For example, to make predictive-

variance-based aleatoric uncertainty measurements more efficient, [24] adds data augmentation during the test-time. But, these works only pay attention to individual uncertainty.

Other recent works attend to the uncertainty measurement for correlated predictive distributions. For example, [29] and [30] measure spatially correlated uncertainty in a generative model for respectively image reconstruction and pixel-wise classification, and [31] captures joint uncertainty as discrete variables in the field of reinforcement learning. Despite these three works, our work is the first to conceptualize and measure collaborative uncertainty in the multi-agent trajectory forecasting task. To the best of our knowledge, there are only two papers [32, 33] close to our track. [32] and [33] model the joint uncertainty in the pose estimation task and multi-agent trajectory forecasting task respectively. However, they present several limitations: 1) They only examined the circumstance where the model's output follows Gaussian distribution; 2) They did not provide a theoretical conceptualization or definition for the uncertainty due to correlated predictive distributions, and they did not analyze the causes of such uncertainty. These are essential problems to tackle. In this work, we not only formulate a general framework that works for both Gaussian and Laplace distributions, but we also theoretically conceptualize *collaborative uncertainty* and analyze its causes.

**Multi-agent trajectory forecasting.** This task takes the observed trajectories from multiple agents as the inputs, and outputs the predicted trajectory for each agent. Like many other sequence prediction tasks, this task used to use a recurrent architecture to process the inputs [34, 35, 36]. Later, however, the graph neural networks become a more common approach as they can significantly assist trajectory forecasting by capturing the interactions among agents [3, 4, 5, 6, 12, 13, 14, 15]. For safety reasons, it is necessary to report the uncertainty of each predicted trajectory. Works to date about uncertainty measurement [11, 21, 22, 37, 38] have appropriately modeled the interaction among multi-agent trajectories for boosting performances, but they overlook the uncertainty resulting from the correlation in predicted trajectories. We seek to fill this gap by introducing and modeling *collaborative uncertainty*.

## 3   Methodology

### 3.1   Problem Formulation

Consider $m$ agents in a data sample, and let $\mathrm{X} = \{\mathbf{x}_1, \mathbf{x}_2, ..., \mathbf{x}_m\}$, $\mathrm{Y} = \{\mathbf{y}_1, \mathbf{y}_2, ..., \mathbf{y}_m\}$ be the past observed and the future trajectories of all agents, where $\mathbf{x}_i \in \mathbb{R}^{2T_-}$ and $\mathbf{y}_i \in \mathbb{R}^{2T_+}$ are the past observed and the future trajectories of the $i$-th agent. Each $\mathbf{x}_i / \mathbf{y}_i$ consists of two-dimensional coordinates at different timestamps of $T_- / T_+$. We assume that a training dataset $\mathcal{D}$ consists of $N$ individual and identically distributed data samples $\{(\mathrm{X}^i, \mathrm{Y}^i)\}_{i=1}^N$. For predicting future trajectories of multiple agents and modeling the uncertainty over the predictions, we seek to use a probabilistic framework to model the predictive distribution $p(\mathrm{Y} \mid \mathrm{X})$ of multiple agents' future trajectories based on the training dataset $\mathcal{D}$. Previous works in uncertainty modeling [16, 17, 39] use Gaussian distribution to approximate $p(\mathrm{Y} \mid \mathrm{X})$.

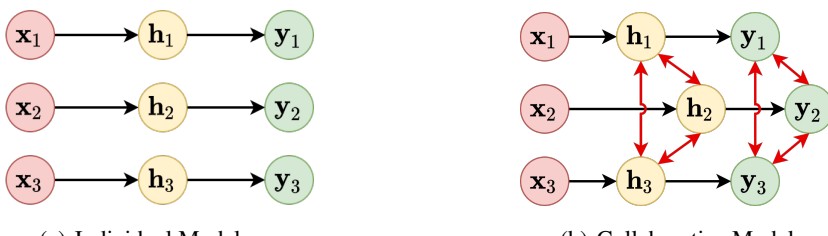

(a) Individual Model          (b) Collaborative Model

Figure 2: **Graphical model for deep learning networks in the three-agent trajectory forecasting setting**: (a) represents the model that predicts the trajectory of each agent independently; (b) shows the model that explicitly captures the interaction among multiple agents. $\mathbf{x}_i$ is the observed trajectory of the $i$-th agent; $\mathbf{h}_i$ and $\mathbf{y}_i$ are its corresponding hidden feature and future trajectory respectively.

The assumption behind this approach is that $p(\mathbf{y}_i | \mathbf{x}_i)$ is independent for every $i \in \{1, 2, 3, ..., m\}$. Mathematically, they set the covariance matrix of $p(\mathrm{Y} \mid \mathrm{X})$ as a diagonal matrix. This assumption is valid for the regression task that uses the model shown in Figure 2(a). We refer the uncertainty under the independence assumption as *individual uncertainty* in this paper. However, Figure 2(b) considers a prediction model that includes interaction modeling among multiple agents: $\mathbf{y}_i$ is no longer dependent solely on $\mathbf{x}_i$, but also on other agents $\mathbf{x}_j$ where $j \neq i$ in the scene. We call the

uncertainty brought by this interaction *collaborative uncertainty*. The existence of collaborative uncertainty turns $p(Y\,|\,X)$ from the individual distribution into the joint distribution of multiple agents.

Contrary to existing methods, we consider collaborative uncertainty and model $p(Y\,|\,X)$ more accurately by making the covariance matrix a full matrix without imposing any restrictions on its form. In the following subsection, we will introduce an approach to modeling both individual uncertainty and collaborative uncertainty using a unified CU-based framework.

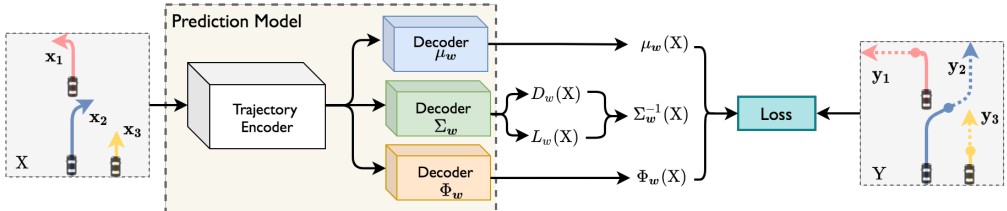

Figure 3: **Proposed uncertainty estimation framework.** The encoder may contain a module that exploits agent-wise interaction. Decoders output the mean $\mu_{\boldsymbol{w}}(X)$, covariance $\Sigma_{\boldsymbol{w}}(X)$ containing individual and collaborative uncertainty, and auxiliary parameters $\Phi_{\boldsymbol{w}}(X)$. The outputs formulate the training loss with the ground truth Y. $\Phi_{\boldsymbol{w}}(\cdot)$ is only used in Laplace collaborative uncertainty.

## 3.2 General Formulation of Collaborative Uncertainty

In this work, to model collaborative uncertainty, we abandon the independence assumption held by previous works [16, 17, 39], setting $p(Y\,|\,X)$ as a joint multivariate distribution, whose mean is $\mu \in \mathbb{R}^{m \times 2T_+}$ and covariance $\Sigma \in \mathbb{R}^{m \times m \times 2T_+}$. Element $\mu_{i,t}$ is the expected position of the $i$-th agent at timestamp $t$. As the diagonal elements of $\Sigma$ are considered individual uncertainty [16, 17, 39], we further let *off-diagonal* elements describe collaborative uncertainty. Diagonal element $\Sigma_{i,i,t}$ models the variance of the $i$-th agent at timestamp $t$; off-diagonal element $\Sigma_{i,j,t}$ models the covariance between the $i$-th and $j$-th agents at timestamp $t$. Therefore, we can simultaneously obtain individual and collaborative uncertainty by estimating the covariance $\Sigma$ of $p(Y\,|\,X)$. Accordingly, we propose a CU-based comprehensive uncertainty estimation framework (see Figure 3) with the following steps:

**Step 1:** *Choose a probability density function,* $p(Y\,|\,X; \mu, \Sigma, \Phi)$, *for the predictive distribution,* which includes a mean $\mu \in \mathbb{R}^{m \times 2T_+}$ used to approximate the future trajectories, a covariance $\Sigma \in \mathbb{R}^{m \times m \times 2T_+}$ used to quantify individual uncertainty and collaborative uncertainty, and some auxiliary parameters $\Phi$ used to describe the predictive distribution. Further, we set covariance matrix $\Sigma_t$, which represents the covariance matrix at timestamp $t$, as a full matrix instead of an identity or diagonal matrix.

**Step 2:** *Design a prediction model,* $\mathcal{F}[\mu_{\boldsymbol{w}}(X), \Sigma_{\boldsymbol{w}}(X), \Phi_{\boldsymbol{w}}(X)]$, where $\mu_{\boldsymbol{w}}(X)$, $\Sigma_{\boldsymbol{w}}(X)$ and $\Phi_{\boldsymbol{w}}(X)$ are three neural networks, which approximate values of mean $\mu$, covariance $\Sigma$ and auxiliary parameters $\Phi$ respectively. Note that $\boldsymbol{w}$ only indicates the parameters of these neural networks are trainable, and does not mean they share same parameters.

**Step 3:** *Derive a loss function* from $p(Y\,|\,X; \mu, \Sigma, \Phi)$ via maximum likelihood estimation: $\mathcal{L}(\boldsymbol{w}) = -\sum\limits_{i=1}^{N} \log p(Y^i\,|\,X^i; \mu_{\boldsymbol{w}}(X^i), \Sigma_{\boldsymbol{w}}(X^i), \Phi_{\boldsymbol{w}}(X^i))$ minimized to update trainable parameters in $\mu_{\boldsymbol{w}}(\cdot)$, $\Sigma_{\boldsymbol{w}}(\cdot)$ and $\Phi_{\boldsymbol{w}}(\cdot)$.

## 3.3 Two Special Cases

Table 1: Two special cases with various assumptions about covariance $\Sigma$. **DIA**: the diagonal matrix (individual uncertainty only). **FULL**: the full matrix (both individual and collaborative uncertainty).

| ASSUMPTION | | LOSS FUNCTION OF TWO SPECIAL CASES | |
|---|---|---|---|
| | | GAUSSIAN DISTRIBUTION | LAPLACE DISTRIBUTION |
| DIA: | $\begin{pmatrix} \sigma_{11} & 0 & \cdots & 0 \\ 0 & \sigma_{22} & \cdots & 0 \\ \vdots & \vdots & & \vdots \\ 0 & 0 & \cdots & \sigma_{mm} \end{pmatrix}$ | $\frac{1}{2}\sum\limits_{i=1}^{m}[\sigma_{ii}^{-2}\|\mathbf{y}_i - \mu_{\boldsymbol{w}}(\mathbf{x}_i)\|_2^2 + \log\sigma_{ii}^2]$ | $\sum\limits_{i=1}^{m}[\sigma_{ii}^{-2}\|\mathbf{y}_i - \mu_{\boldsymbol{w}}(\mathbf{x}_i)\|_1 + \log\sigma_{ii}^2]$ |
| FULL: | $\begin{pmatrix} \sigma_{11} & \sigma_{12} & \cdots & \sigma_{1m} \\ \sigma_{21} & \sigma_{22} & \cdots & \sigma_{2m} \\ \vdots & \vdots & & \vdots \\ \sigma_{m1} & \sigma_{m2} & \cdots & \sigma_{mm} \end{pmatrix}$ | $\frac{1}{2}[q_{\boldsymbol{w}}(Y,X) - \sum\limits_{j=1}^{m}\log(d_{jj})]$ | $\frac{1}{2}[\frac{q_{\boldsymbol{w}}(Y,X)}{\Phi_{\boldsymbol{w}}(X)} + m\log\Phi_{\boldsymbol{w}}(X) - \sum\limits_{j=1}^{m}\log(d_{jj})]$ |

In multi-agent trajectory forecasting, based on Laplace and Gaussian distributions, the $\ell_1$- and $\ell_2$-based loss functions are commonly adopted to train prediction models [3, 4, 15, 40]. Here we apply the probabilistic framework proposed in Section 3.2 to model the individual and collaborative uncertainty based on multivariate Gaussian distribution and multivariate Laplace distribution respectively, which leads to two novel loss functions. Mathematically, the essential difference between our proposed loss functions and previous loss functions derived from Gaussian distribution and Laplace distribution for modeling individual uncertainty is that they have different assumptions about the covariance matrix; see a summary in Table 1. We regard the covariance as a full matrix.

### 3.3.1 Gaussian collaborative uncertainty

We start by the multivariate Gaussian distribution, as it has a simpler probability density function than the multivariate Laplace distribution.

**Probability density function.** We follow the framework proposed in Section 3.2 and choose the probability density function as the multivariate Gaussian distribution:

$$p(\mathrm{Y}\,|\,\mathrm{X}; \mu, \Sigma, \Phi) = (2\pi)^{-\frac{m}{2}} \cdot \det[\Sigma]^{-\frac{1}{2}} \cdot e^{-\frac{1}{2}(\mathrm{Y}-\mu)\Sigma^{-1}(\mathrm{Y}-\mu)^T}, \tag{1}$$

where $\det[\Sigma]$ represents the determinant of covariance $\Sigma$.

**Model design.** Based on (1), we can approximate the value of mean $\mu$ via a neural network $\mu_{\boldsymbol{w}}(\cdot)$. When using the same way to approximate the value of covariance $\Sigma$, however, we face two challenges: 1) each covariance matrix $\Sigma_t$ in covariance $\Sigma$ needs to be inverted, which could lead to numerical instability; 2) it is computationally expensive and numerically unstable to compute the determinant of each covariance matrix $\Sigma_t$ in covariance $\Sigma$ directly given a large amount of trainable parameters.

For the first challenge, we use a neural network $\Sigma_{\boldsymbol{w}}^{-1}(\cdot)$ to directly approximate the inverse of covariance $\Sigma$. For the second challenge, similar to [29] and [32], we apply the square-root-free Cholesky decomposition to each $\Sigma_{t_{\boldsymbol{w}}}^{-1}$ in $\Sigma_{\boldsymbol{w}}^{-1}(\mathrm{X})$: $\Sigma_{\boldsymbol{w}}^{-1}(\mathrm{X}) = L_{\boldsymbol{w}}(\mathrm{X})D_{\boldsymbol{w}}(\mathrm{X})L_{\boldsymbol{w}}^T(\mathrm{X})$, where $L_{\boldsymbol{w}}(\mathrm{X})$ is a lower unit triangular matrix and $D_{\boldsymbol{w}}(\mathrm{X})$ is a diagonal matrix. Then, the determinant of the inverse of covariance $\Sigma^{-1}$ is obtained by $\prod_{j=1}^{m} d_{jj}$, where $d_{jj}$ is the $j$-th diagonal element in $D_{\boldsymbol{w}}(\mathrm{X})$.

We can thus get the parameterized form of (1) as: $p(\mathrm{Y}|\mathrm{X};\boldsymbol{w}) = (2\pi)^{-\frac{m}{2}} (\prod_{j=1}^{m} d_{jj})^{\frac{1}{2}} e^{-\frac{q_{\boldsymbol{w}}(\mathrm{Y},\mathrm{X})}{2}}$, where $q_{\boldsymbol{w}}(\mathrm{Y},\mathrm{X}) = [\mathrm{Y}-\mu_{\boldsymbol{w}}(\mathrm{X})]\Sigma_{\boldsymbol{w}}^{-1}(\mathrm{X})[\mathrm{Y}-\mu_{\boldsymbol{w}}(\mathrm{X})]^T$.

As there are no auxiliary parameters in the parameterized form of (1), we can get the prediction model $\mathcal{F}[\mu_{\boldsymbol{w}}(\mathrm{X}), \Sigma_{\boldsymbol{w}}^{-1}(\mathrm{X})]$, whose framework is illustrated in Figure 3. Once $\Sigma_{\boldsymbol{w}}^{-1}(\mathrm{X})$ is fixed and given, individual and collaborative uncertainty are computed through the inversion.

**Loss function.** According to the square-root-free Cholesky decomposition and the parameterized form of (1), the Gaussian collaborative uncertainty loss function is then:

$$\mathcal{L}_{\mathrm{Gau-cu}}(\boldsymbol{w}) = \frac{1}{2}\frac{1}{N}\sum_{i=1}^{N}[q_{\boldsymbol{w}}(\mathrm{Y}^i, \mathrm{X}^i) - \sum_{j=1}^{m} \log(d_{jj}^i)]. \tag{2}$$

We update the trainable parameters in $\mu_{\boldsymbol{w}}(\cdot)$ and $\Sigma_{\boldsymbol{w}}^{-1}(\cdot)$ through minimizing (2). Note that $q_{\boldsymbol{w}}(\cdot,\cdot)$ is related to $\mu_{\boldsymbol{w}}(\cdot)$ and $\Sigma_{\boldsymbol{w}}^{-1}(\cdot)$, and $d_{jj}^i$ is related to $\Sigma_{\boldsymbol{w}}^{-1}(\cdot)$.

### 3.3.2 Laplace collaborative uncertainty

In multi-agent trajectory forecasting, previous methods [3, 4, 15] have found that the $\ell_1$-based loss function derived from Laplace distribution usually leads to better prediction performances than the $\ell_2$-based loss function from Gaussian distribution, because the former is more robust to outliers. It is thus important to consider multivariate Laplace distribution.

**Probability density function.** We follow the framework proposed in Section 3.2 and choose the probability density function as the multivariate Laplace distribution:

$$p(\mathrm{Y}\,|\,\mathrm{X}; \mu, \Sigma, \Phi) = \frac{2\det[\Sigma]^{-\frac{1}{2}}}{(2\pi)^{\frac{m}{2}}\lambda} \cdot \frac{K_{(\frac{m}{2}-1)}(\sqrt{\frac{2}{\lambda}(\mathrm{Y}-\mu)\Sigma^{-1}(\mathrm{Y}-\mu)^T})}{(\sqrt{\frac{\lambda}{2}(\mathrm{Y}-\mu)\Sigma^{-1}(\mathrm{Y}-\mu)^T})^{\frac{m}{2}}}, \tag{3}$$

where $\det[\Sigma]$ denotes the determinant of covariance $\Sigma$, and $K_{(\frac{m}{2}-1)}(\cdot)$ denotes the modified Bessel function of the second kind with order $(\frac{m}{2}-1)$.

**Model design.** Similar to Section 3.3.1, we employ two neural networks $\mu_{\boldsymbol{w}}(\cdot)$ and $\Sigma_{\boldsymbol{w}}^{-1}(\cdot)$ to approximate the values of $\mu$ and $\Sigma^{-1}$ respectively, and represent $\Sigma_{\boldsymbol{w}}^{-1}(X)$ via its square-root-free Cholesky decomposition we used in the Gaussian collaborative uncertainty. Since the modified Bessel function is intractable for a neural network to work with, different from Section 3.3.1, we should simplify (3).

Inspired by [41], we simplify (3) by utilizing the multivariate Gaussian distribution to approximate the multivariate Laplace Distribution. We reformulate a multivariate Laplace distribution by introducing auxiliary variables. Let $\mathbf{z} \in \mathbb{R}^+$ be a random variable with the probability density function: $p(\mathbf{z} | X; \boldsymbol{w}) = \frac{1}{\lambda} e^{-\frac{\mathbf{z}}{\lambda}}$, then we can get: $p(Y | \mathbf{z}, X; \boldsymbol{w}) = \frac{\det[\Sigma_{\boldsymbol{w}}^{-1}(X)]^{\frac{1}{2}}}{(2\pi\mathbf{z})^{\frac{m}{2}}} e^{-\frac{q_{\boldsymbol{w}}(Y,X)}{2\mathbf{z}}}$, where $q_{\boldsymbol{w}}(Y, X) = [Y - \mu_{\boldsymbol{w}}(X)]\Sigma_{\boldsymbol{w}}^{-1}(X)[Y - \mu_{\boldsymbol{w}}(X)]^T$. Further, if the value of $\mathbf{z}$ is given, $p(Y | \mathbf{z}, X; \boldsymbol{w})$ is a multivariate Gaussian distribution. In this work, instead of drawing a value for $\mathbf{z}$ from the exponential distribution, we use a neural network $\Phi_{\boldsymbol{w}}(\cdot)$ to directly output a value for $\mathbf{z}$. The intuition is that, in the training process of the prediction model, the value of $p(Y | X; \boldsymbol{w})$ is the conditional expectation of $\mathbf{z}$ given X and Y, which makes $p(Y | \mathbf{z}, X; w)$ a function of $\mathbf{z}$ whose domain is $\mathbb{R}^+$. Thus, there should exist an appropriate $\mathbf{z}^* \in \mathbb{R}^+$ to make: $p(Y | X; \boldsymbol{w}) = p(Y | \mathbf{z}^*, X; \boldsymbol{w})$ (see proof in the appendix). To find such a $\mathbf{z}^*$, we use $\Phi_{\boldsymbol{w}}(X)$, which can employ its learning ability. Then, we can get the parameterized form of $p(Y | X; \boldsymbol{w})$ as: $p(Y | X; \boldsymbol{w}) = \frac{\det[\Sigma_{\boldsymbol{w}}^{-1}(X)]^{\frac{1}{2}}}{(2\pi\Phi_{\boldsymbol{w}}(X))^{\frac{m}{2}}} e^{-\frac{q_{\boldsymbol{w}}(Y,X)}{2\Phi_{\boldsymbol{w}}(X)}}$.

Finally, we can get the prediction model $\mathcal{F}[\mu_{\boldsymbol{w}}(X), \Sigma_{\boldsymbol{w}}^{-1}(X), \Phi_{\boldsymbol{w}}(X)]$, whose framework is illustrated in Figure 3. Individual and collaborative uncertainty are indirectly learned by the $\Sigma_{\boldsymbol{w}}^{-1}(X)$.

**Loss function.** On the basis of the square-root-free Cholesky decomposition and the parameterized form of $p(Y | X; \boldsymbol{w})$, the Laplace collaborative uncertainty loss function is then:

$$\mathcal{L}_{\text{Lap-cu}}(\boldsymbol{w}) = \frac{1}{2} \frac{1}{N} \sum_{i=1}^{N} \left[ \frac{q_{\boldsymbol{w}}(Y^i, X^i)}{\Phi_{\boldsymbol{w}}(X^i)} + m \log \Phi_{\boldsymbol{w}}(X^i) - \sum_{j=1}^{m} \log(d_{jj}^i) \right]. \tag{4}$$

where $d_{jj}^i$ is the $j$-th diagonal element in $D_{\boldsymbol{w}}(X^i)$. The parameters of $\mu_{\boldsymbol{w}}(\cdot)$, $\Sigma_{\boldsymbol{w}}^{-1}(\cdot)$ and $\Phi_{\boldsymbol{w}}(\cdot)$ are updated by minimizing (4). And $q_{\boldsymbol{w}}(\cdot, \cdot)$ is related to $\mu_{\boldsymbol{w}}(\cdot)$ and $\Sigma_{\boldsymbol{w}}^{-1}(\cdot)$, and $d_{jj}^i$ belongs to $\Sigma_{\boldsymbol{w}}^{-1}(\cdot)$.

### 3.4 Discussion

After presenting how to quantify collaborative uncertainty in Section 3.2 and Section 3.3, here we discuss the nature of collaborative uncertainty.

As mentioned in Section 3.2, we can divide a prediction model for multi-agent trajectory forecasting into two types: individual models and collaborative models. An individual model predicts the future trajectory and the corresponding uncertainty for each agent independently; a collaborative model leverages an interaction module to explicitly capture the interactions among multiple agents, which makes all the predicted trajectories correlated. Moreover, this interaction modeling procedure can bring extra uncertainty to the model; in other words, we consider that the interaction modeling in a prediction model leads to collaborative uncertainty (CU).

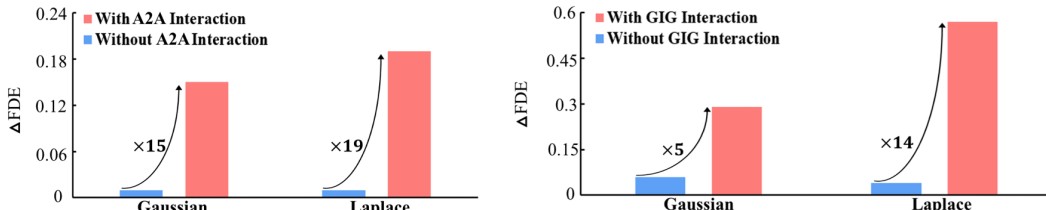

Figure 4: **CU modeling gains more when a prediction model includes interaction modeling.** Blue and red bars reflect the gains from CU modeling for a prediction model w/o an interaction module respectively. $\Delta$FDE is the difference of FDE values between versions w/o CU modeling.

To validate this, we can empirically compare how much impact of collaborative uncertainty modeling would bring to an individual model and a collaborative model. In our experiment, we use two cutting-edge multi-agent trajectory forecasting models, LaneGCN [3] and VectorNet [4].[2] In

---

[2]To focus on agents' behaviors, we remove the map information and the related modules in this experiment.

LaneGCN/VectorNet, an agent-to-agent (A2A) module/a global-interaction-graph (GIG) module explicitly models the interactions among multiple agents. As illustrated in Figure 4, when we remove the A2A/GIG module from LaneGCN/VectorNet, collaborative uncertainty modeling brings much less gain to LaneGCN/VectorNet; see Section 4.2.3 for more details of this experiment. This reflects that the cause of collaborative uncertainty mainly comes from the interaction modeling.

# 4   Experiments

We first use two self-generated synthetic datasets with a limited number of agents as the toy version of the real world problem. We use the simplified datasets to test our method's ability of capturing distribution information of the input data that obeys a certain type of multivariate distribution. We then conduct extensive experiments on two published benchmarks to prove the value of our proposed method in solving real world problems. We introduce the experiments in Sec. 4.1 and Sec. 4.2.

## 4.1   Toy Problem

We define a toy problem to validate the capability of the proposed framework for accurate probability distribution estimation. The toy problem requires models to take the mutually correlated trajectories sampled from a given distribution as the input and output the mean and the covariance of the distribution.

As far as we know, in real-world datasets, we only have the ground truth for the predicted trajectory, which is the mean of the distribution while have no access to the ground truth of the uncertainty, which is the covariance matrix of the distribution. Therefore, we generate two synthetic datasets with the ground truth for both mean and covariance matrix given two distributions respectively.

**Datasets.** We generate two synthetic datasets for ternary Gaussian and ternary Laplace distribution respectively. Each dataset contains training, validation and test sets, which have 36000, 7000 and 7000 instances respectively. Each instance includes the trajectories of three agents, which consist of the two-dimensional point coordinates of the three agents at 50 different timestamp. In each instance, the trajectories of three agents are sampled from a ternary joint Gaussian/Laplace distribution. Generation details are provided in the appendix.

**Implementation details.** The network architecture used in the experiment contains an encoder and two decoders, all of which are four-layer MLPs. The neural network outputs the predicted mean and covariance matrix of the given ternary Gaussian/Laplace distribution. Although the ground truth covariance matrix is known for synthetic datasets, they are not used in training. We train the network using the previous uncertainty modeling method and our proposed method on each synthetic dataset.

**Metric.** Here, we adopt three metrics for evaluation: the $\ell_2$ distances between the estimated mean and the ground truth mean, the $\ell_1$ distances between the estimated covariance matrix and the ground truth covariance matrix and the KL divergence between the ground truth distribution and the estimated distribution. We provide metrics computing details in the appendix.

**Evaluation results.** The test set results of the synthetic dataset are shown in Table 2 and Figure 5. Since the previous uncertainty modeling method only models IU (individual uncertainty) but our method models both IU and CU (collborative uncertainty), our method allows the model to estimate a more accurate mean and covariance matrix on a given distribution, which leads to a much lower KL divergence between the ground truth distribution and the estimated distribution.

## 4.2   Real World Problem

The experiments results above show the superiority of our method for simplified problems. Here we further validate its ability on two real-world large-scale autonomous driving datasets in single future prediction. We also conduct ablation studies in this subsection to present the effectiveness of the method design.

### 4.2.1   Experiment setup

**Datasets.** *Argoverse* [42] and *nuScenes* [47] are two widely used multi-agent trajectory forecasting benchmarks. *Argoverse* has over 30K scenarios collected in Pittsburgh and Miami. Each scenario is a sequence of frames sampled at 10 Hz. The sequences are split as training, validation and test sets, which have 205942, 39472 and 78143 sequences respectively. *nuScenes* collects 1000 scenes in Boston and Singapore. Each scene is annotated at 2 Hz and is 20s long. The prediction instances are split as training, validation and test sets, which have 32186, 8560 and 9041 instances respectively.

Table 2: Comparison with the prior uncertainty modeling method on synthetic datasets under the two different assumptions of distribution. $\mu_{\boldsymbol{w}}(X)$: the estimated mean. $\mu_{gt}$: the ground truth mean. $\Sigma_{\boldsymbol{w}}(X)$: the estimated covariance matrix. $\Sigma_{gt}$: the ground truth covariance matrix. KL: the KL divergence $D_{KL}(p_g(X)\|p_e(X))$, where $p_e(X)$ is the estimated distribution and $p_g(X)$ is the ground truth distribution.

|  | GAUSSIAN | | LAPLACE | |
|---|---|---|---|---|
|  | IU ONLY | IU + CU | IU ONLY | IU + CU |
| $\ell_2$ OF $\mu$ | 0.68 | **0.49** | 0.42 | **0.34** |
| $\ell_1$ OF $\Sigma$ | 1.98 | **1.01** | 1.96 | **1.13** |
| KL | 6.68 | **0.40** | 12.6 | **1.65** |

Table 3: Comparison with SOTA methods on Argoverse test set. CU boosts performances in single future prediction.

| METHOD | ADE | FDE |
|---|---|---|
| ARGO BASELINE (NN) [42] | 3.45 | 7.88 |
| ARGO BASELINE [42] | 2.96 | 6.81 |
| UULM-MRM [43, 44] | 1.90 | 4.19 |
| VECTORNET [4] | 1.81 | 4.01 |
| TNT [45] | 1.78 | 3.91 |
| JEAN [46] | 1.74 | 4.24 |
| LANEGCN [3] | 1.71 | 3.78 |
| OURS: | | |
| LANEGCN (OUR IMPLEMENTATION) | 1.76 | 3.84 |
| USING $\mathcal{L}_{\text{Gau-cu}}$ | 1.73 | 3.83 |
| USING $\mathcal{L}_{\text{Lap-cu}}$ | **1.70** | **3.74** |

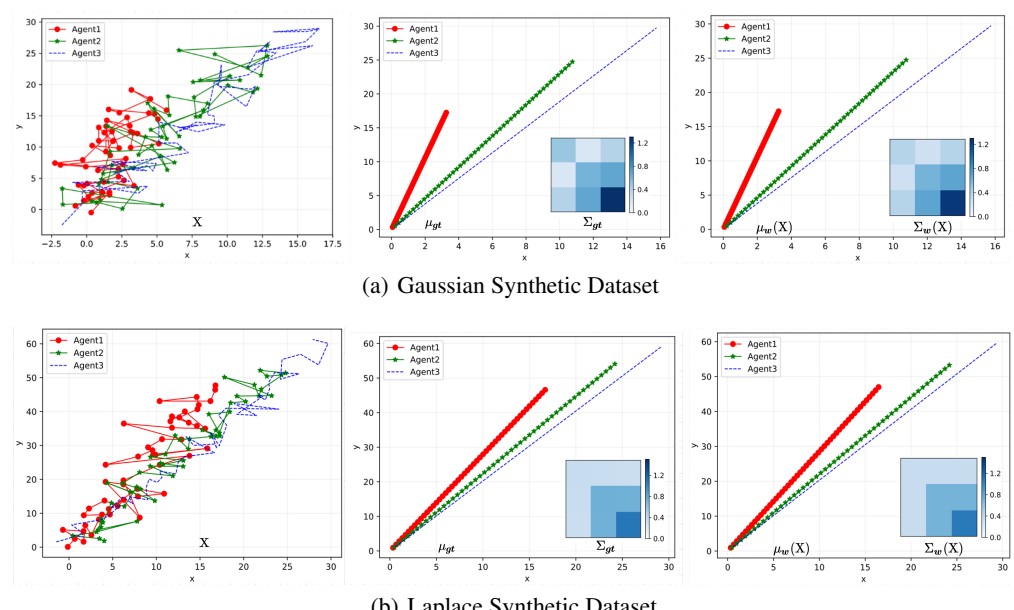

(a) Gaussian Synthetic Dataset

(b) Laplace Synthetic Dataset

Figure 5: **Sample visualization on synthetic dataset.** Our proposed CU-based framework allows the model to learn the mean and covariance matrix of ground truth distribution accurately. Each input instance X is the numerical sum of a ground truth mean $\mu_{gt}$ and a random variable $\epsilon$ with the information of the ground truth covariance matrix $\Sigma_{gt}$. Please find more details in Appendix.

For Argoverse, we forecast future trajectories for 3s based on the observed trajectories of 2s. For nuScenes, we forecast future trajectories for 6s based on the observed trajectories of 2s.

**Metrics.** We adopt two extensively used multi-agent trajectory forecasting metrics, Average Displacement Error (ADE) and Final Displacement Error (FDE). ADE is defined as the average of pointwise $\ell_2$ distances between the predicted trajectory and ground truth. FDE is defined as the $\ell_2$ distance between the final points of the prediction and ground truth (ADE and FDE are measured in meters).

**Implementation details.** The proposed model is implemented follow the basis of LaneGCN [3] and VectorNet [4], two cutting-edge multi-agent trajectory forecasting models. We implement the encoder of LaneGCN/VectorNet as the trajectory encoder in our proposed model. We further use the four-layer multilayer perceptrons (MLPs) to respectively implement three decoders $\mu_{\boldsymbol{w}}(\cdot)$, $\Sigma_{\boldsymbol{w}}(\cdot)$ and $\Phi_{\boldsymbol{w}}(\cdot)$. Note that all experiments are based on our own implementation of LaneGCN/VectorNet, which may not perform as well as the official LaneGCN/VectorNet.

Table 4: Ablation on covariance $\Sigma$ with chosen probability density function (**PDF**) and Interaction module (**INT.**). **DIA** denotes the diagonal matrix (individual uncertainty). **FULL** denotes the full matrix (individual and collaborative uncertainty). On Argoverse and nuScenes, both LaneGCN and VectorNet with individual and collaborative uncertainty surpasses the ones with individual uncertainty only. Collaborative uncertainty makes a larger impact for the model with an interaction module.

| METHOD | DATASET | PDF TYPE | INT. | ADE DIA | ADE FULL | $\Delta$ADE | FDE DIA | FDE FULL | $\Delta$FDE |
|---|---|---|---|---|---|---|---|---|---|
| LANEGCN | ARGOVERSE | GAUSSIAN | × | 1.69 | 1.67 | 0.02 | 3.88 | 3.85 | 0.03 |
| | | | √ | 1.45 | 1.42 | **0.03** | 3.19 | 3.14 | **0.05** |
| | | LAPLACE | × | 1.67 | 1.67 | 0.00 | 3.82 | 3.82 | 0.00 |
| | | | √ | 1.43 | **1.41** | 0.02 | 3.16 | **3.11** | 0.05 |
| | NUSCENES | GAUSSIAN | × | 4.60 | 4.58 | 0.02 | 11.02 | 11.01 | 0.01 |
| | | | √ | 4.47 | 4.39 | 0.08 | 10.59 | 10.44 | 0.15 |
| | | LAPLACE | × | 4.53 | 4.52 | 0.01 | 10.93 | 10.92 | 0.01 |
| | | | √ | 4.34 | **4.25** | 0.09 | 10.34 | **10.15** | 0.19 |
| VECTORNET | ARGOVERSE | GAUSSIAN | × | 1.82 | 1.78 | 0.04 | 4.16 | 4.08 | 0.08 |
| | | | √ | 1.63 | 1.57 | **0.06** | 3.60 | 3.46 | **0.14** |
| | | LAPLACE | × | 1.78 | 1.76 | 0.02 | 4.06 | 4.02 | 0.04 |
| | | | √ | 1.56 | **1.52** | 0.04 | 3.42 | **3.34** | 0.08 |
| | NUSCENES | GAUSSIAN | × | 4.25 | 4.23 | 0.02 | 10.35 | 10.29 | 0.06 |
| | | | √ | 4.07 | 3.99 | 0.08 | 9.86 | 9.57 | 0.29 |
| | | LAPLACE | × | 4.19 | 4.18 | 0.01 | 10.25 | 10.21 | 0.04 |
| | | | √ | 4.02 | **3.81** | 0.21 | 9.79 | **9.22** | 0.57 |

#### 4.2.2 Results

**Evaluation results on benchmark datasets.** In this subsection, we implement our proposed framework based on LaneGCN [3] as it is the SOTA model in multi-agent trajectory forecasting. We compare our proposed methods in the Argoverse trajectory forecasting benchmark with two official baselines, including Argoverse Baseline [42] and Argoverse Baseline (NN) [42], and five SOTA methods of this benchmark: LaneGCN [3], TNT [45], Jean [46], VectorNet [4] and uulm-mrm [43, 44]. Table 3 shows that although our implementation of LaneGCN appears less good than the official implementation of LaneGCN, our implementation of LaneGCN with the individual and collaborative uncertainty notably outperforms all of the other competing methods in both ADE and FDE. Therefore, our proposed collaborative uncertainty modeling enhances the SOTA prediction models.

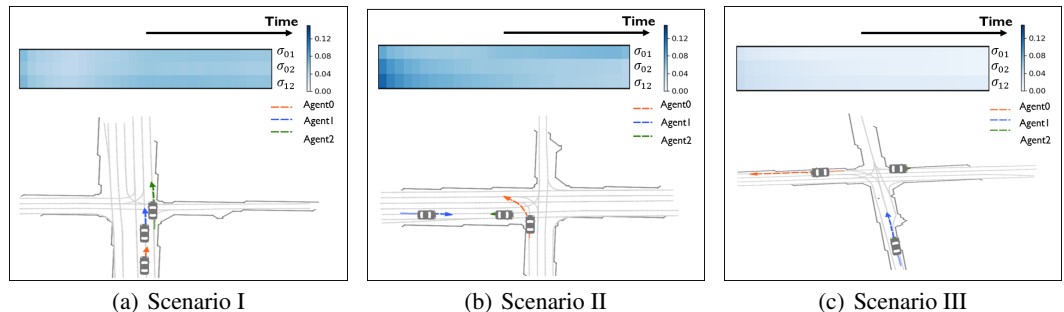

| (a) Scenario I | (b) Scenario II | (c) Scenario III |
|---|---|---|

Figure 6: **Visualization of CU on Argoverse dataset**. (a) In scenario I, Agent0 and Agent1 are driving on the same road, which is next to the road where Agent2 is driving. This type of scenario might generate complicated interactive information making $\sigma_{01}$, $\sigma_{02}$ and $\sigma_{12}$ show a non-monotonic change over time. (b) In scenario II, Agent0 and Agent1 are moving towards each other, thus $\sigma_{01}$ increases over time. Agent2 is parking at an intersection waiting for green light, as little new interactive information between Agent2 and the other two agents would be generated before the red light turns green, $\sigma_{02}$ and $\sigma_{12}$ decrease over time. (c) In scenario III, Agent0, Agent1 and Agent2 are located in completely different areas heading to different directions, $\sigma_{01}$, $\sigma_{02}$ and $\sigma_{12}$ are close to zero.

**Visualization of collaborative uncertainty.**[3] To visually understand which factor might influence the value of collaborative uncertainty in multi-agent trajectory forecasting, we present the visualization results generated by our model in Figure 6. We visualize 3 scenarios, and each scenario includes 3 agents (orange, blue and green lines) trajectories (solid lines are the past trajectories and dashed lines are the future trajectories) and their corresponding collaborative uncertainty values changing over the last 30 frames (the heatmap, where $\sigma_{ij}$ denotes the collaborative uncertainty of agent $i$ and agent $j$). These results show that the value of collaborative uncertainty is highly related to the amount of the interactive information among agents.

### 4.2.3   Ablation study

We study: 1) how different approaches of uncertainty modeling would affect the prediction model; 2) how the interaction module would influence the effectiveness of collaborative uncertainty modeling. In this part, we adopt LaneGCN/VectorNet as our trajectory encoder for proving that our proposed method can be used as a plug-in module to improve the performances of existing models in multi-agent trajectory forecasting. To focus on the agents' behaviors, we ignore map information and map-related modules in LaneGCN/VectorNet; we only use the agent encoder of LaneGCN/VectorNet, which extracts intermediate feature from agents' past trajectories and actor-to-actor (A2A) module/global-interaction-graph (GIG) module that exploits the interactions among agents. In this part, the experiments are conducted on the validate sets of Argoverse and nuScenes benchmarks.

**Different approaches of uncertainty modeling.** We consider two approaches of uncertainty modeling: assuming the covariance matrix is a diagonal matrix (DIA) to model individual uncertainty only, and assuming the covariance matrix is a full matrix (FULL) to model both individual and collaborative uncertainty. Based on the results in Table 4, we see that 1) modeling individual and collaborative uncertainty together (FULL) is consistently better than modeling individual uncertainty only (DIA); and 2) our proposed Laplace CU-based framework enables LaneGCN and VectorNet to achieve the best performances on ADE & FDE metrics on both Argoverse and nuScenes benchmarks. These results reflect that our proposed collaborative uncertainty modeling can work as a plugin module to significantly improve the prediction performance of existing models.

**Effects of interaction module.** A2A/GIG module is the interaction module in LaneGCN/VectorNet. We study the effects of the interaction module in collaborative uncertainty modeling via the ablation study on A2A/GIG module. ∆ADE/FDE is the difference of ADE/FDE values between versions with and without collaborative uncertainty. Higher values reflect bigger gains brought by modeling collaborative uncertainty. From Table 4, gains from modeling collaborative uncertainty in a collaborative model (with A2A/GIG) are much greater than in an individual model (without A2A/GIG). Figure 4 visualizes the ∆FDE on nuScenes benchmark; see discussions in Section 3.4.

## 5   Conclusions

This work proposes a novel probabilistic collaborative uncertainty (CU)-based framework for multi-agent trajectory forecasting. Its key novelty is conceptualizing and modeling CU introduced by interaction modeling. The experimental results in this work demonstrate the ability of the proposed CU-based framework to boost the performance of SOTA prediction systems, especially collaborative-model-based systems. The proposed framework potentially lead to more reliable self-driving systems.

This work shows the promise of our method for single-future trajectory prediction, which predicts the single best trajectory for agents. With growing research interests in predicting multiple potential trajectories for agents (i.e. multi-future trajectory prediction), we leave the generalization of our method to multi-future trajectory prediction as an important direction for future research.

## Acknowledgement

This work was supported in part by the National Key R&D Program of China under Grant 2021YFB1714800, the National Natural Science Foundation of China under Grant 6217010074, 62173034, 62088101, the Science and Technology Commission of Shanghai Municipal under Grant 21511100900, the DiDi GAIA Research Collaboration Plan, and the CAAI-Huawei MINDSPORE Academic Award Fund.

---

[3]These visualization results are based on Gaussian CU. Please see more visualization results in the appendix.

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
