# OpenReview forum: "Collaborative Uncertainty in Multi-Agent Trajectory Forecasting"
_NeurIPS.cc/2021/Conference — NeurIPS 2021 Poster_

### Official Review · Reviewer_vRWU · 2021-07-16

**Rating:** 6
**Confidence:** 4

**Summary:**

The paper presents a novel concept — collaborative uncertainty — in the context of multi-agent trajectory forecasting. Unlike many previous works that only model the individual uncertainty of joint future trajectory distribution and use a diagonal covariance matrix, this paper considers both individual and collaborative uncertainty of multi-agent trajectories and uses a full covariance matrix with off-diagonal components representing collaborative uncertainty between agents. The paper argues that such a design could model the GT multi-agent trajectory distribution more accurately. The paper performs synthetic experiments showcasing the method's ability to model joint trajectory distribution better than models that only consider individual uncertainty. Experiments on real-world trajectory forecasting benchmarks also demonstrate that incorporating collaborative uncertainty improves the performance of deterministic baselines.

**Limitations And Societal Impact:**

- The paper discussed its limitations, i.e., currently only applied to deterministic trajectory forecasting models.
- The paper didn't discuss its societal impact, as deemed not applicable by the authors.

**Main Review:**

**Strength:**

- The proposed concept of collaborative uncertainty is novel in the context of multi-agent trajectory forecasting. Although there are works that model the joint distribution of multi-agent trajectories, little work has addressed the uncertainty (covariance) stemming from the interaction of different agents. Collaborative uncertainty could become an important tool for measuring interactions between agents and improve the interpretability of trajectory forecasting models. I believe this is also a new direction other people could build on.
- The experiments demonstrated the effectiveness of collaborative uncertainty modeling as it improves modeling synthetic data distributions as well as improving the performance of existing deterministic trajectory forecasting methods. It is also nice to see that collaborative uncertainty mainly improves models with interaction modeling, confirming that the performance gain is coming from better uncertainty modeling in agent interaction.
- The paper is generally well-written and well-motivated and super easy to understand.
- The paper also presents detailed derivations of the non-trivial case of Laplace distribution and addresses several practical problems of how to approximate the distribution efficiently and provides proof to justify predicting the auxiliary parameters z.

**Weakness:**

- As admitted by the authors, the biggest limitation of this paper is that collaborative uncertainty has only been applied to deterministic trajectory forecasting models. It is unclear how difficult it is to apply it to generative models such as VAEs and GANs. It is also unclear how effective the method would be for those generative models as these models already have stochasticity and uncertainty built in by latent variables. Given the popularity and effectiveness of generative trajectory forecasting methods, many of them also consider stochastic interaction modeling, the contributions of this paper diminish when not applying to those generative models.
- The method is technically a generative model, and the collaborative uncertainty seems to be able to deal with stochasticity in agent interaction. Thus, it would be good to generate multiple samples and compare with other generative models on both Argoverse and nuScenes datasets for metrics like minADE_k and minFDE_k. The current FDE_1 and ADE_1 results are not very competitive on the nuScenes leaderboard, and many methods with better FDE_1 exist, such as [a]. It would be better to apply collaborative uncertainty to these methods.
- The visualization in Fig 6 only shows simple interaction scenarios when there are only two cars and many scenarios with little or no interaction. It would be more interesting to show the interaction of multiple agents (≥ 3) and how their collaborative uncertainty changes over time. I also don't understand why in Pair 1, the collaborative uncertainty becomes lower as time goes by. One would expect the collaborative uncertainty goes higher as agents are closer and have more chances to interact.

[a] Trajectory Prediction for Autonomous Driving based on Multi-Head Attention with Joint Agent-Map Representation. Messaoud et al. 2020.

**Minor Comments:**

Typos:

- line 105:  change "," to "." or use "and" after ","
- line 257: "shows" → "show"

**Time Spent Reviewing:**

5

---

> ### Author Response · Authors · 2021-08-09
> **Response to Reviewer vRWU's comments**
>
> First of all, thank you very much for your time and your valuable comments. We fixed the typos and improved the visualization of the figures in the revised manuscript according to your feedbacks.
>
> *Q1. Why is this method not applied to the generative model?*
>
> R1. We evaluated this option and decided to apply this method only to the deterministic model for two reasons.
>
> 1) Compared to the deterministic model, the generative model would bring about a larger and more complex influence both on and from the uncertainty model, which would make our collaborative uncertainty analysis less straightforward. For instance, if applied to the generative model, we would set both p(z|x) and p(y|z) as the multivariate Gaussian/ multivariate Laplace distributions with full covariance matrix. In this way, there would be two covariance matrices, one from p(z|x) and one from p(y|z). As a result, a novel mathematical framework and model structure would be required to measure the collaborative uncertainty from these two covariance matrices, which will introduce confounding to our proposed method. **As the objective of this paper is to introduce a novel concept of collaborative uncertainty to the general readership, we believe a more simplistic model would more robustly and straightforwardly convey the concept.**
>
> 2) We treat the application of collaborative uncertainty to the generative model as a part of our future work towards multi-future trajectory prediction. Many previous works [1][2] have shown that the generative models are suitable for multi-future trajectory prediction. Since we consider multi-future trajectory prediction as the future work, we do not specifically study the generative model in the current work. Please kindly find the reason why we did not study multi-future trajectory prediction in this work in our 3rd response (R3) to Reviewer 9D3B.
>
> ***
>
> *Q2. The current $FDE_1$ and $ADE_1$ results are not very competitive on the nuScenes leaderboard, and many methods with better $FDE_1$ exist, such as [a]. It would be better to apply collaborative uncertainty to these methods. [a] Trajectory Prediction for Autonomous Driving based on Multi-Head Attention with Joint Agent-Map Representation. Messaoud et al. 2020.*
>
> R2. Thank you for the suggestion! We did not test our method on other models except for LaneGCN[3] and Vectornet[4] from the nuScenes leaderboard mainly for two reasons:
>
> 1) We consider the main purpose of this work to be establishing a new concept, collaborative uncertainty, and evaluating its impact on models in multi-agent trajectory forecasting. For this purpose, we chose two well-known methods: LaneGCN[3] and Vectornet[4], which have been published in the top conferences in this area and have been substantially replicated and validated. We believe their representativeness can warrant the generalizability of our method.
>
> 2) The main purpose of our work is not to get the best perform on the nuScenes leaderboard. We did the ablation study on the nuScenes dataset in order to comprehensively study the benefits that collaborative uncertainty brings to models in multi-agent trajectory forecasting. Therefore, we did not specifically choose the models from the nuScenes leaderboard.
>
> On the other hand, we agree with you that it would be nice if our ideas are tested on more methods from the nuScenes leaderboard. We read Paper [a] you mentioned, and indeed found very relevant and inspiring technical contents. However, **this paper's referred github repository is empty. Without any open source code, the limited rebuttal period does not permit us to implement their method from scratch.** We will try our best to reproduce the paper you recommended and show the results in the revised manuscript.
>
> ***
>
> *Q3. Qualitative results for scenarios with multiple agents ($\geq 3$) are needed.*
>
> R3. Indeed, the qualitative results including multiple agents ($\geq 3$) will be interesting. We only show qualitative results with only two agents because we want to use a simplistic interaction type to make the work more comprehensible to the general readership, especially in understanding the variation of collaborative uncertainty in a given period. We will follow your advice and add qualitative results for scenarios with multiple agents to the revised manuscript. Thank you!
>
> ***
>
> *Q4. In Pair 1 of Fig 6, why does the collaborative uncertainty become lower as time goes by?*
>
> R4. This might be caused by the fact that, in pair 1, one agent is approaching another agent, which is stopped at an intersection and waiting for green light. Even these two agents are geographically closer, little new interactive information would be generated before the red light turns green.
>
> ***
>
> *Q5. What's the societal impact of this method?*
>
> R5. We believe this method would impose minimal to none negative impacts to the society, whereas its positive societal impact is to make self-driving systems more reliable. We will add a few sentences about the societal impact of this method to the revised manuscript. Thank you!
>
> [1]Y. Yuan and K. Kitani. Diverse trajectory forecasting with determinantal point processes. ICLR, 2020.
>
> [2]T. Salzmann, B. Ivanovic, P. Chakravarty, and M. Pavone, “Trajectron++: Dynamically-feasible trajectory forecasting with heterogeneous data,” in ECCV, 2020.
>
> [3]Ming Liang, Bin Yang, Rui Hu, Yun Chen, Renjie Liao, Song Feng, and Raquel Urtasun. Learning lane graph representations for motion forecasting. In ECCV, 2020.
>
> [4]Jiyang Gao, Chen Sun, Hang Zhao, Yi Shen, Dragomir Anguelov, Congcong Li, and Cordelia Schmid. Vectornet: Encoding hd maps and agent dynamics from vectorized representation. In Proceedings of the IEEE/CVF Conference on Computer Vision and Pattern Recognition (CVPR), June 2020.

---

> > ### Comment · Reviewer_vRWU · 2021-08-31
> > **Update**
> >
> > After reading the response, I have decided to keep my current score. The response has not addressed my concerns about comparisons with generative models and analysis of multiple agents (>= 3). I generally think the concept proposed by the paper is interesting, although I have some doubt about its usefulness compared with generative models. I wouldn't mind seeing this paper accepted but will not fight for it.

---

### Official Review · Reviewer_q6Rk · 2021-07-17

**Rating:** 8
**Confidence:** 3

**Summary:**

The work introduces the idea of estimating joint uncertainties between all actors in a motion forecasting setting, instead of the established way of doing so by actor individually. The concept is simple enough to be applied to a variety of different existing motion forecasting systems and the authors demonstrate that the additional loss leads to better prediction accuracy after training.

Overall I liked this paper a lot. It may be a simple idea but it is powerful, as the authors demonstrate in their results. The paper is also very clearly written and the experiments are sufficient.

**Limitations And Societal Impact:**

I think you missed the part of the authors' instructions that said to discuss limitations and societal impact of this method. I have to slightly lower my overall score of the paper to be fair to other submissions that discussed these in detail.

For societal impact, I'd mention that this doesn't have any impact beyond other trajectory forecasting models and that the big possible positive consequence could be having more reliable self-driving cars.

For limitations, I'd mention for example that it might not be practical to calculate the predicted CU between all actors in a scene in crowded scenes. Specifically, in a dense urban environment with many pedestrians, this might be tricky and maybe some heuristic might be required here. Also, I'd like to see more of the safety aspect implemented, i.e. the abstract mentioned how this was great for making systems more explainable but then this wasn't really executed.

**Main Review:**

**Originality:**

Estimating single-agent uncertainty is quite common but joint uncertainty or "collaborative uncertainty" is less well-studied. In model-based RL, works like [1] try to capture joint uncertainty as discrete variables but this does not address the uncertainty over the motion of individual actors. The only work that comes to mind, that has attempted something like this in motion forecasting is the recent [2]. In this work, the model captures both a per-actor uncertainty at every step and the collaborative uncertainty, which is expressed as discrete modes that can condition the future rollouts.
I still think this work is sufficiently novel since it's a more general (theoretical) framework but it may be good to briefly mention this recent work in the related works section.

**Quality:**

The submission is technically very sound. Well done!

**Clarity:**

The submission is clearly written and well-organized. I liked the little toy example before the main application to Argoverse and nuScenes. I wasn't able to follow all the math in section 3.3.1 but I hope another reviewer can help point out if this makes sense or not. The rest of the paper is clear.
Together with the supplementary video, Fig.6 is great but on its own, it's harder to parse. Maybe a link in the caption of the figure to the video would be great for the final version of the paper.
I also liked the detailed depictions of the networks and explanation of training details in the appendix, making this easy to reproduce.

**Significance:**

I think this method is straightforward and easy to adapt into a multitude of motion forecasting systems, leading to better predictive results and more explainability.

**Nitpicks and Questions:**

1. Starting in the abstract, you write multiple times that something approximates something "nicely" and I think that's a bit too colloquial for a paper
2. line 15: rebuild -> approximate?
3. line 1: no dash between trajectory and forecasting
4. contribution 2 in the abstract mentions improving predictive performance but up until then, only safety+explainability was mentioned
5. Last sentence of abstract sounds like "our predicted output is highly related to the input"
6. Fig. 1 is great but Fig. 2 is kind of redundant wrt Fig. 1. Maybe it'd be possible to redesign Fig. 1 to be a bit bigger and bolder like Fig. 2 and then cut Fig. 2?
7. What's the complexity to compute the loss in eq 4 over the variant with only the diagonal covariance matrix?
8. To read+review, I printed the paper in black/white and Fig. 5a is utterly unreadable that way. But even looking at the color image in the PDF, I can hardly make out what's happening there.
9. Section 4.2.1 is great and more papers should do a quick recap like this.
10. How do you explain the vastly different $\Delta ADE$s across the same model and the same dataset in Tab. 4?
11. Why do you think the Laplacian model leads so consistently to better ADE results in Tab.4?
12. line 285 "less good" -> "worse"
13. line 301 "plug" -> "plugin"? "plug & play"?

**References:**

- [1] Hafner, Danijar, et al. "Mastering atari with discrete world models." arXiv preprint arXiv:2010.02193 (2020).
- [2] Girgis, Roger, et al. "Latent Variable Nested Set Transformers & AutoBots." arXiv preprint arXiv:2104.00563 (2021).

**Time Spent Reviewing:**

4

---

> ### Author Response · Authors · 2021-08-09
> **Response to Reviewer q6Rk's comments**
>
> Thank you very much for your kind comments and constructive feedback! We improved the writing and the quality of the figures in the revised manuscript according to your suggestions.
>
> *Q1.  About two papers[1][2] mentioned in the comment.*
>
> R1. Thank you very much for mentioning these two interesting works! We will add some contents about these two works in the related works. The first paper [1] mainly introduce a novel model DreamerV2 in the field of reinforcement learning. This proposed model mentions a global exploration based on uncertainty estimates. The second one [2] proposes a new class of Latent Variable Sequential Set Transformers in the field of multi-agent trajectory prediction. This work models the uncertainty at every step. These works also inspire us the potential application of the proposed collaborative uncertainty towards the deep reinforcement learning framework. The main difference between our work and these two works is that, as you mentioned, we formulate a general framework, which works for both Gaussian and Laplace distributions, and theoretically conceptualize collaborative uncertainty and analyze its causes. Again, some contents about these two works will be added in the related works. Thanks!
>
> ***
>
> *Q2. What's the complexity to compute the loss in Eq 4 over the variant with only the diagonal covariance matrix?*
>
> R2. Over the variant with only the diagonal covariance matrix, the complexity to compute the loss in Eq 4 is the same as that to compute the loss with only IU, which is $O(N)$. This is because we only need to use a $N$-dimensional vector to represent the diagonal elements of the covariance matrix.
>
> ***
>
> *Q3. About the Fig.5a.*
>
> R3. The Fig.5a is an illustration of one of the instances of the input data in the toy problem. In the toy problem, each instance of the input data includes the trajectories of three agents, which consist of the two-dimensional point coordinates of the three agents at 50 different timestamp. These trajectories of three agents are sampled from a ternary joint Gaussian/Laplace distribution. In the Fig.5a, the three broken lines in different colors are the trajectories of three different agents of an instance of the input data. Sorry for the inconvenience to the readers of the printed version, we will further clarify the caption of this figure and use three different styles of lines to represent the trajectories. Thank you for this clarification question!
>
> ***
>
> *Q4. How do you explain the vastly different $\Delta$ADEs across the same model and the same dataset in Tab. 4?*
>
> R4. We assume that by "vastly different" you meant that the $\Delta$ADEs had a larger difference in Vectornet with interaction modules on the nuScenes dataset between Gaussian (0.08) and Laplace (0.21) PDF types. This is the only difference that we think might be considered relatively large (0.13), as the other differences in $\Delta$ADEs between the two PDF types across the same model and the same dataset in Tab. 4 were relatively small, between 0.01 and 0.02. Thus, we consider the large difference in  $\Delta$ADEs as a special case instead of a significant and consistent pattern.
>
> ***
>
> *Q5. Why do you think the Laplacian model leads so consistently to better ADE results in Tab.4?*
>
> R5. In our opinion, the Laplacian model leads consistently to better ADE results because of the different characteristics of Gaussian and Laplace distributions as well as the signal to noise ratio (SNR) of the datasets.
>
> 1) The Gaussian model may be more susceptible to noises in the data than the Laplacian model. Compared with Gaussian distribution, the energy of Laplace distribution is more concentrated in the center of the distribution, which is the position of the mean of the distribution. Such different characters of the Gaussian distribution and the Laplace distribution make the former more sensitive to outliers than the latter, causing the Gaussian model more likely to sacrifice many normal samples for adapting to single outliers.
>
> 2) Both Argoverse and nuScenes are datasets obtained from the real world traffic scenes through the sensors on some vehicles. They inevitably contain both environmental noises from the real world scenes and equipment noises introduced by the sensors. Subsequently, the SNR in Argoverse and nuScenes can be relatively low.
>
> Since the Gaussian model is more vulnerable to the noises and these two datasets with relative low SNR, the Laplacian model consistently outperforms the Gaussian model.
>
> ***
>
> *Q6. What's the limitations and the societal impact of this method?*
>
> R6. Regarding the limitations, we have discussed some of them in the last paragraph of the conclusion section. A main limitation of our work is the lack of the generalization to multi-future trajectory prediction. The main reason for this limitation is that we think, in the context of multi-future trajectory prediction, the collaborative uncertainty analysis will be less straightforward and the accessibility of this new concept of collaborative uncertainty will be largely decreased. Because multi-future trajectory conditions would introduce confounding variables and require heavier modification to existing models. For instance, for the multi-future trajectory prediction, it is important for us to make the predicted trajectories diverse. In our opinion, there should be some relationships between the diversity and the uncertainty of the predicted trajectories (e.g., the scenarios with high uncertainty should also have high diversity). As a result, for using the uncertainty in multi-future trajectory prediction, we have to design a delicate framework to handle the relationships between the diversity and the uncertainty. Please kindly find more reasons for the existence of this limitation in the 3rd response (R3) to Reviewer 9D3B. Therefore, because there is a lot of work need to do for applying the uncertainty to multi-future trajectory prediction, we treat it as one of the future works can be built based on this work and in this paper we only discuss collaborative uncertainty in the context of single future prediction. We will further clarify this limitation and add some more limitations based on all three reviewers' valuable feedback.
>
> Regarding the the societal impacts, we think this method would impose minimal to none negative impacts to the society, whereas its positive societal impact, as you nicely put, is to make self-driving systems more reliable. We will add a few sentences about the societal impact of this method to the revised manuscript. Thank you!
>
> [1]Hafner, Danijar, et al. "Mastering atari with discrete world models." arXiv preprint arXiv:2010.02193 (2020).
>
> [2]Girgis, Roger, et al. "Latent Variable Nested Set Transformers \& AutoBots." arXiv preprint arXiv:2104.00563 (2021).

---

> > ### Comment · Reviewer_q6Rk · 2021-08-17
> > **Thanks**
> >
> > Thanks for the clarifications. And re "Q/R4" - yeah I just looked back into it and I can't figure out what I actually intended to say there. Please disregard that point.

---

### Official Review · Reviewer_9D3B · 2021-07-20

**Rating:** 6
**Confidence:** 3

**Summary:**

** Update 8/19 **

Thanks to the authors for providing detailed responses to comments. Based on their clarifications and the other reviews, I have updated my score.

**

The authors propose a collaborative uncertainty (CU) module to model the uncertainty between agents for multi-agent trajectory prediction.
It is evidenced that when the CU module is incorporated into two recently published models without environment information, there is an improvement in the predictive accuracy.
The authors also provide a visualization for CU, which shows that in situations where there should be no collaborative uncertainty, CU does not produce unintuitive results.


**Limitations And Societal Impact:**


For improvement suggestions, please refer to the full review.


**Main Review:**

The references are appropriate, and the exposition has good clarity.
Please also cite these works:
	Ye, Maosheng, Tongyi Cao, and Qifeng Chen. "TPCN: Temporal Point Cloud Networks for Motion Forecasting." Proceedings of the IEEE/CVF Conference on Computer Vision and Pattern Recognition. 2021.
	Gilles, Thomas, et al. "HOME: Heatmap Output for future Motion Estimation." arXiv preprint arXiv:2105.10968 (2021).

Quantitative Experiments:
The quantitative experiments include comparisons to the recent works and leverage well-studied evaluation metrics.
However, I have several concerns.
1.	The authors do not report the number of predicted trajectories used for computing ADE and FDE in any of their results tables. I believe it is K=1.
2.	It is very common for the compared models to rely on the min ADE/FDE across 6 predicted trajectories per test case instead of just 1. This manuscript only reports the results for K=1. As such, it is unclear whether the improvement from the CU module is observed in these critical metrics (min ADE and min FDE for K=6).
3.	Table 3 reports the ADE/FDE for Jean with K=1 from the public Argoverse leaderboard, which conflicts with the Jean results reported in the LaneGCN and VectorNet papers.
4.	The results reported for TNT conflict with the public Argoverse leaderboard.
5.	It is unclear whether the CU module would improve the performance of the current SOTA: HOME, LaneRCNN, and TPCN, which all outperform LaneGCN according to K=1. These should be included in Table 3 and the application of CU should be discussed w.r.t. these works as well.

The toy problem shows simple cases in which the proposed collaborative uncertainty module performs well.
Since the random variable epsilon is effectively noise and independent between synthetic agents, it appears that the synthetic agent trajectories are completely non-interacting (e.g., there are no collisions). Yet, according to Figure 5, there is collaborative uncertainty. Shouldn't this result match Pairs 5-8 in Figure 6?
It is also unclear how this toy problem relates to the trajectories of real interacting agents.

Qualitative results are not shown for scenarios with more than 3 interacting agents in the naive toy problem or more than 2 interacting agents in the Argoverse dataset.
It is therefore difficult to evaluate the estimated CU for important situations with many interacting agents.


**Time Spent Reviewing:**

2 hours

---

> ### Author Response · Authors · 2021-08-09
> **Response to Reviewer 9D3B 's comments**
>
> Thank you very much for your constructive feedbacks. We added more clarification in the revised manuscript according to your suggestions.
>
> *Q1. Please also cite these works: Ye, Maosheng, Tongyi Cao, and Qifeng Chen. "TPCN: Temporal Point Cloud Networks for Motion Forecasting." Proceedings of the IEEE/CVF Conference on Computer Vision and Pattern Recognition. 2021. Gilles, Thomas, et al. "HOME: Heatmap Output for future Motion Estimation." arXiv preprint arXiv:2105.10968 (2021).*
>
> R1. We added the suggested references. Thank you!
>
> ***
>
> *Q2. The authors do not report the number of predicted trajectories used for computing ADE and FDE in any of their results tables. I believe it is K=1.*
>
> R2. Yes, it is K=1 for predicted trajectories in computing ADE and FDE. We will clarify this in the results. Thanks!
>
> ***
>
> *Q3. This manuscript only reports the results for K=1. As such, it is unclear whether the improvement from the CU module is observed in these critical metrics (min ADE and min FDE for K=6).*
>
> R3. We agree that comparing models across 6 or more predicted trajectories would be interesting. But we believe the setting of single trajectory prediction (K=1) is more compact and self-contained to achieve our main purpose, which is to establish the new concept, collaborative uncertainty, and evaluate its impact on the prediction models.
>
> The reasons are as followings:
>
> 1) The setting of multi-future trajectory prediction would hurt the accessibility of this new concept of collaborative uncertainty. Because the setting of multi-future trajectory prediction would introduce many confounding variables (e.g., the diversity of multiple predicted trajectories) and require heavier modification (e.g., some new mechanisms used to balance the uncertainty and the diversity) to existing models. Therefore, this would make our collaborative uncertainty analysis less straightforward and robust.
>
> 2) The setting of multi-future trajectory prediction would cause an additional challenge of handling the relationships between the diversity and the uncertainty (either individual uncertainty or collaborative uncertainty). For multi-future trajectory prediction, it is important to make the predicted trajectories diverse in an appropriate manner. There would be some connection between the diversity and the uncertainty of the predicted trajectories (e.g., the scenarios with high uncertainty should also have high diversity). Therefore, to leverage the uncertainty in multi-future trajectory prediction, we need an additional delicate framework to handle the relationships between diversity and uncertainty, which might cause the distraction from our core concept, collaborative uncertainty.
>
> 3) The setting of multi-future trajectory prediction would cause an additional challenge of handling the relationships between confidence score and uncertainty (either individual uncertainty or collaborative uncertainty). In existing multi-future prediction works, their models usually generate a confidence score to represent the credibility of each predicted trajectory, which is used for selecting the best predicted trajectory. Intuitively, the uncertainty might carry the similar spirit with a confidence score. But how to leverage the uncertainty for the trajectory selection is still an open issue, requiring further dedicated designs. We consider such designs as an application of the uncertainty and do not strongly relate to our novel concept, collaborative uncertainty.
>
> 4) The results of single-future trajectory prediction are sufficient and self-contained to validate the effectiveness and practicality of the proposed collaborative uncertainty. In our work, we test our proposed framework based on two important SOTAs (VectorNet and LaneGCN). VectorNet and LaneGCN have been substantially replicated and validated. Their representativeness warrants the generalizability of our method.
>
> As shown above, **there is a lot of work to do for applying the uncertainty (either individual or collaborative) to improve the prediction performance of  multi-future trajectory prediction, and predicting multi-future trajectories seems relatively digressive to our central goal of expounding the proposed collaborative uncertainty.** Therefore, we regard it as a future work that may be completed on the basis of collaborative uncertainty, which our current work focuses on.
>
> ***
>
> *Q4. Table 3 reports the ADE/FDE for Jean with K=1 from the public Argoverse leaderboard, which conflicts with the Jean results reported in the LaneGCN and VectorNet papers. The results reported for TNT conflict with the public Argoverse leaderboard.*
>
> R4. Such a confusion results from the fact that the data on the leaderboard is continuously being updated as people submit new results. We got the results of Jean and TNT from the public Argoverse Motion Forecasting Competition leaderboard on Feb. 4, 2021. You could find the same leaderboard results in Table 1 of "Wenyuan Zeng, Ming Liang, Renjie Liao, and Raquel Urtasun. LaneRCNN: Distributed representations for graph-centric motion forecasting. arXiv preprint arXiv:2101.06653, 2021". **This matching should reflect that our results are precise and trustworthy.**
>
> ***
>
> *Q5. It is unclear whether the CU module would improve the performance of the current SOTA: HOME, LaneRCNN, and TPCN, which all outperform LaneGCN according to K=1. These should be included in Table 3 and the application of CU should be discussed w.r.t. these works as well.*
>
> R5. Thank you for the suggestion. We agree with you on the value of HOME, LaneRCNN, and TPCN. **To the best of our knowledge, however, none of these three methods offers open source codes.** "HOME: Heatmap Output for future Motion Estimation" and "TPCN: Temporal Point Cloud Networks for Motion Forecasting" were accepted by CVPR 2021 after we submitted our work to NeurIPS 2021. Without open source codes, the current time and resources do not permit us to implement three more methods from scratch. In our work, we test our proposed framework based on two important SOTAs (VectorNet and LaneGCN). VectorNet and LaneGCN have been substantially replicated and validated. Their representativeness warrants the generalizability of our method.
>
> ***
>
> *Q6. How does this toy problem relate to the trajectories of real interacting agents? And why does CU exist in the scenario of toy problem?*
>
> R6. To answer the first question, we regard the simulated toy problem as a supplement to the real world problem to validate the mathematical rationality of our proposed method. As this paper proposes a probabilistic framework, we need to validate that the proposed framework can accurately learn the mean and the covariance matrix of a trajectory's distribution. However, in real-world problems, we only have the ground truth for the predicted trajectory, which is the mean of the distribution, and do not have the ground truth for the uncertainty, which is the covariance matrix of the distribution. Therefore, we design a toy problem owning the synthetic datasets with the ground truth for both mean and covariance matrix of the given distribution. In the toy problem, we generate two synthetic datasets for ternary Gaussian and ternary Laplace distribution respectively, and the models are required to learn the mean and covariance of the distribution $p(\mathbf{x})$ of the given dataset based on the input data $\mathbf{x}$.
>
> To answer the second question, according to the illustrations of Sections 1 and 3 in our paper, the collaborative uncertainty can be mathematically represented as the covariance of the trajectory distribution, which is the ideal output of a prediction model. As, in the toy problem, the models aim at learning the distribution $p(\mathbf{x})$ of the given dataset, we think the covariance of $p(\mathbf{x})$ coincides the mathematical representation of collaborative uncertainty.
>
> We will refine the writing about the toy problem so that readers may better understand the role of the toy problem in our paper. Thank you for this clarification question!
>
> ***
>
> *Q7. Qualitative results are not shown for scenarios with more than 3 interacting agents.*
>
> R7. Indeed, it will be interesting to visualize more complex interactions. Originally, qualitative results are shown for scenarios with only two agents because we believe that a simplistic interaction type will help make the work more comprehensible to the general readership, especially in understanding the variation of collaborative uncertainty in a given period. We will follow your advice and add more qualitative results in the revised manuscript.

---

> ### Comment · Area_Chair_tMBd · 2021-08-19
> **Towards reviewer agreement**
>
> Hi,
>
> Just a reminder to please look over the author responses and let your fellow reviewers and the ACs know if you still feel this work is below threshold.
>
> Thanks,
> The AC

---

> > ### Comment · Reviewer_9D3B · 2021-08-19
> > **Updated review score**
> >
> > I have updated my score based on the rebuttal and other reviews.
> > I have no objections for this paper to be accepted if others are willing to champion the paper.

---

### Public Comment · ~Michael_A._Alcorn1 · 2021-11-17
**Concurrent Relevant Work - baller2vec++**

Congratulations on your paper being accepted to NeurIPS! I wanted to bring your attention to [our manuscript](https://arxiv.org/abs/2104.11980), which was concurrently submitted to NeurIPS but was unfortunately [rejected](https://openreview.net/forum?id=p2XgjS3Qp4X). In our manuscript, we address the same problem of temporal statistical dependencies between agents, but our approach uses an interleaved input design combined with a Transformer, which allows the model to learn the full joint distribution of the agents' trajectories by exploiting the chain rule of probability. Anyway, I was wondering if you could clarify some aspects of your model for me?

First, could you provide more details on how the MLPs generate the covariance matrices? The outputs of VectorNet and LaneGCN are graphs, so it wasn't clear to me how you were going from a graph to a covariance matrix. Figure 1 gives the impression that the agent representations are concatenated together, but how was the order of the agents determined in that case? In `baller2vec++`, we handled the lack of a natural order in the agents by randomly permuting the order of the agents during training.

Second, you state that the shape of Σ is m × m × 2T, which makes it sound like there are separate covariance matrices for each dimension of the trajectory (i.e., x and y). Is that correct? I would've expected the shape of Σ to be 2m × 2m × T. Otherwise, I think there could be issues in the generated trajectories under certain conditions. E.g., if two agents move approximately along the slope y = x approximately in unison, the covariance matrices for x and y will be similar, and many of the generated trajectories given these covariance matrices will be unrealistic (see my code snippet [here](https://gist.github.com/airalcorn2/206fea0ad3384a9ec68e05d0f8f67a60#file-multimodal_multivariate-py-L9) for a visualization).

Third, the multivariate Gaussian/Laplacian loss is assuming the agents' trajectories are unimodal, right? This also seems potentially problematic under certain scenarios. For example, imagine that you have two 1D agents who, 50% of the time, move in unison one unit to the right (with some noise), and the other 50% of the time move in unison one unit to the left. The true joint distribution of their trajectories will have two modes at (1, 1) and (-1, -1). If you assume a multivariate normal distribution for these trajectories, the maximum likelihood estimate for the mean (which is equal to the mode for a multivariate normal distribution) is (0, 0), which translates to neither agent moving, i.e., the region containing the estimated mean effectively *never* occurs in the dataset. Further, the maximum likelihood estimate for the covariance matrix will have ones for the covariance terms, which leads to the learned distribution for the trajectories looking quite different from the true distribution (see my code snippet [here](https://gist.github.com/airalcorn2/206fea0ad3384a9ec68e05d0f8f67a60#file-multimodal_multivariate-py-L43) for visualizations). I think the chain rule factorization in `baller2vec++` might make it easier to learn arbitrary distributions (in fact, we demonstrate `baller2vec++`'s ability to learn multimodal distributions with a toy dataset).

Lastly, have you thought about how your approach could be applied to categorical outputs (e.g., actions)? This is another case where I think the chain rule factorization in `baller2vec++` might make things easier.

---

### Decision · Program_Chairs · 2021-09-27

**Decision:**

Accept (Poster)

**Comment:**

All reviewers recommend accepting this paper.

The work addresses the issue of estimating joint uncertainties between actors in a motion forecasting setting.
Experiments are provided on some commonly used benchmarks for motion forecasting.
Reviewers generally felt the paper was clearly written and the experiments were sufficient.

The AC recommends acceptance.